# Experimental Studies on Chemical Activation of Cementitious Materials from Smelting Slag of Copper and Nickel Mine

**DOI:** 10.3390/ma12020303

**Published:** 2019-01-18

**Authors:** Lijuan Wang, Yanke Wei, Guocheng Lv, Libing Liao, Dan Zhang

**Affiliations:** 1Beijing Key Laboratory of Materials Utilization of Nonmetallic Minerals and Solid Wastes, National Laboratory of Mineral Materials, School of Materials Science and Technology, China University of Geosciences, Beijing 100083, China; wljcl@cugb.edu.cn (L.W.); wei13140165707@163.com (Y.W.); 2Beijing General Research Institute of Mining & Metallurgy, Beijing 110160, China

**Keywords:** smelting slag of copper and nickel deposits, gellable materials, chemical activation, sodium sulfate

## Abstract

Gellable composite materials (GCM) were prepared from a smelting slag of copper and nickel deposits and cement, and activated using gypsum and chemical activators. The effects of material ratio, dosage of chemical activators, and gypsum on the mechanical properties of GCM were studied. Our results showed that the chemical activators of Na_2_SO_4_, Na_2_SiO_3_, NaOH, and Na_2_CO_3_ could improve the compressive strength of the GCM. Considering the market cost and ease operation, the compressive strength of the GCM could be significantly improved with 2% Na_2_SO_4_. The experiment results also showed that the compound chemical activator could improve the compressive strength of gelled material. The strength of GCM reaches 41.6 MPa when 2% gypsum and 80% of smelting slags of copper and nickel deposits were used, which met the national standards requirements of GCM. As such, it is expected that a large amount of copper and nickel mining smelting slag could be utilized for the production of cementitious materials.

## 1. Introduction

In the process of utilizing mineral resources, a series of environmental problems could arise, such as production of larger amounts of waste rock and gob, waste of resources, and security risks. The stacking of tailings and smelting slag from metallurgical process produced a serious pollution to the environment. Filling the gobs with tailings and other solid waste can not only control solid waste pollution and improve the utilization of the land, but also prevent the surface subsidence and improve the mining index. However, if tailings and other solid waste are directly used to fill the gobs, they will be subject to chemical oxidation and ion release under rain, sun, and other natural stresses.

Nonferrous smelting slag is the waste residue in the smelting process of nonferrous metals and minerals [1,2]. China is the largest non-ferrous smelting country in the world [3]. At present, the slag of the non-ferrous smelting industry reaches 128 million tons in China [4], and its comprehensive utilization is 22.4 million tons [5], accounting for only 17.5%, which is far lower than that of developed countries. The nonferrous smelting industry has made a great contribution to China’s economic development, but also caused serious environmental problems, like occupation of land, and pollution of the environment and groundwater [6,7,8]. Filling technology was developed rapidly as an effective way to overcome the mining problems in the past 20 years [9,10,11]. On the other hand, fundamental scientific research is very important and urgent to improve the use of filling mineral resources and to solve the stability of current mining filling material and high processing cost [12,13,14].

An effective way to use nonferrous smelting slag is to prepare filling cementitious material from large amounts of solid waste produced in mining and metallurgy [15,16,17,18]. The low gelling activity and difficulty in stimulating the nonferrous smelting slag are the main factors restricting the large-scale utilization of nonferrous smelting slag in China [19,20,21].

The cementitious material of chemically activated smelting slag is a kind of cementitious material that uses nonferrous smelting slag as raw material and produces the cementitious property through the process of chemical activation [22,23,24,25]. Due to different types of chemical activators and differences in composition and structure of smelting slags, the chemical reaction and hardening process of hydration would be very complex, and the composition and structure of hydration products are quite different [26,27,28,29]. As such, it is difficult to get a unified rule. In this study, the effects of four chemical activators on the smelting slag of copper and nickel deposits (SSCND)–cement system in Xinjiang province were studied [30,31]. Our results provided a best scenario for the comprehensive utilization of SSCND for the production of cementitious materials.

## 2. Material and Methods

### 2.1. Materials

The SSCND was provided by Xinjiang Kalatongke Mining Co., LTD. (Altay, China) Due to their larger particle size, samples were ball-milled for 7 h before being used. Their chemical composition is shown in Table 1.

Portland cement made of silicate cement clinker tricalcium silicate, dicalcium silicate, tricalcium aluminate, and tetracalcium ferroaluminate was used as the benchmark cement. It was provided by China United Cement Group Co., LTD. It has a strength grade higher than 42.5 MPa after grinding with addition of gypsum, whose main component is calcium dihydrate CaSO_4_·2H_2_O.

The chemical activators were sodium hydroxide (NaOH), anhydrous sodium sulfate (Na_2_SO_4_), anhydrous sodium carbonate (Na_2_CO_3_), and sodium silicate (Na_2_SiO_3_·9H_2_O). All were of pure analytical grade.

### 2.2. Characterizations

SSCND at 90%, 80%, 70%, 60%, or 50% was premixed with 10%, 20%, 30%, 40%, or 50% of cement, respectively, for 30 s, according to GB/T17671-1999 “cement mortar strength testing method (ISO) made of cement mortar specimen”, followed by demolding after 24 h standard curing with the mold size of 20 mm × 20 mm × 20 mm, at 23 ± 1 °C and 90% humidity [32]. The specimen was continuously cured under standard conditions for 28 days.

The strength tests were conducted using an electrohydraulic servo pressure test machine under microcomputer control manufactured by Changchun New Testing Machine Co., LTD. (Changchun, China) The main technical parameters included (i) high vacuum mode at resolutions of 1.0 nm-15 kV (TLD-SE, FEI company, Hillsboro, OR, USA), 1.4 nm-1 kV (TLD-SE); (ii) low vacuum mode at resolutions of 1.5 nm-10 kV (HelixTM detector, FEI company, Hillsboro, OR, USA), 1.8 nm-3 kV (HelixTM detector), and (iii) landing voltage of 50~30 kV. The morphological observation was made using the Nova NanoSEM50 series of ultra-high resolution scanning electron microscopy produced by FEI company (Hillsboro, OR, USA). X-ray diffraction (XRD) analyses were performed with CuKα radiation under 40 kV and 100 mA and a scanning speed at 8°/min.

### 2.3. Experiment

Tests for specimen strength were conducted on the 3rd and 28th days [33], and were recorded as T1-X. The ratio of SSCND to cement that meets the required strength was then selected. Different chemical activators were used to activate the SSCND–cement samples with this ratio. The strength of the samples on the 3rd and 28th days was measured, denoted as T2-X. The chemical activator with the best activation effect was then selected. Different amounts of chemical activator were used to activate smelting slag–cement samples, and the strength of the samples at the 3rd and 28th days was measured and denoted as T3-X, to determine the optimal amount of chemical activator. Desulphurization gypsum with different proportions was added to conduct composite activation of SSCND–cement samples with gypsum and chemical activator added, and the strength of the samples at the 3rd and 28th days was measured and recorded as T4-X.

## 3. Results and Discussion

### 3.1. Preparation of GCM from SSCND

For the study of comprehensive performance of SSCND–cement, the experimental conditions listed in Table 2 were used. The samples were premixed for 30 s, and foaming ensured by adding water. Then, the slurry was mixed, and specimens prepared. After 3 and 28 days, the compressive strength was measured [34]. 

The compressive strength of samples with 100% cement and 100% slag on the 28th day reached 78.9 MPa and 2.2 MPa, respectively. Figure 1 showed that when the content of cement increases, the compressive strength increases continuously. This is because cement is a good gelling material [35] and has a good compressive strength after hardening. When the dosage of cement was 50%, the compressive strength of the GCM reached a maximum value of 57.7 MPa after 28 days of curing. When the dosage of cement was 20%, the compressive strength of the gelled material was 22.5 MPa on the 28th day, which would be close to the national standard of 32.5 MPa after chemical activation. Due to the high production cost of cement, chemical activators were added in the following experiments, which would also increase the strength of the GCM. In accordance with the principle of saving production cost and taking full use of smelting slag, T1-2 mixture ratio was adopted as the optimal experimental proportion for subsequent experiments.

### 3.2. The Effects of Different Chemical Activators on the Performance of the GCM

In order to evaluate the role of activation under different chemical activators for GCM, the experimental scheme in Table 3 was used for further experiments. The chemical activators of sodium sulfate, sodium silicate, sodium carbonate, and sodium hydroxide, with a dosage of 6%, were used to activate GCM. Then, the compressive strength was tested, and the best activating reagent confirmed.

The SSCND was premixed with cement for 30 s before water was added to guarantee molding. The slurry was stirred and cured for 3 and 28 days to obtain the experimental samples. The compressive strength of the samples tested is shown in Figure 2.

The compressive strength of GCM activated with 6% Na_2_SO_4_ as an external chemical activator reached 24.2 MPa after 3 days, which is the highest of the four samples. This is because Na_2_SO_4_ was often used as the main components of concrete early-strength agent, which plays an important role in improving the structure and performance of cement concrete. In addition, the compressive strength of samples with NaOH as an additional chemical activator after curing for 3 days reached a value of 16.3 MPa (Figure 2). This is because NaOH is a strong alkali, which has an obvious effect on accelerating the early hydration of cement [36]. The hydration rate of the GCM activated by Na_2_CO_3_ is also faster, which is due to the fact that Na_2_CO_3_ with low incorporation can effectively accelerate the hydration reaction. The hydration rate of Na_2_SiO_3_ is slow. Although the strength of the GCM with Na_2_SO_4_ and NaOH added as the activator for 28 days met the requirements of the national standard (32.5 MPa), NaOH chemical activator has many drawbacks, like high cost, strong alkaline, highly corrosive, low operability, and non-suitability for industrial production [37]. When Na_2_SO_4_ was used as the activator, the compressive strength of the GCM reached the highest and conformed to the strength requirements of the GCM [38,39]. At the same time, Na_2_SO_4_ is low in cost and high in operability, with good actual activation effects. It is a neutral salt that has low corrosion. For comprehensive consideration, Na_2_SO_4_ was selected as the chemical activator.

### 3.3. The Effect of the Dosage of Na_2_SO_4_ on GCM

In order to further study the effect of the dosage of Na_2_SO_4_ on the GCM of SSCND–cement system, 1%, 2%, 3%, 6%, 9%, and 12% Na_2_SO_4_ was used to activate the GCM (Table 4). The compressive strength of the GCM further confirmed the best dosage of Na_2_SO_4_ (Figure 3).

It can be illustrated that when the added amount of Na_2_SO_4_ was between 1% and 6%, the compressive strength for the 3rd and 28th days increased. At 6% Na_2_SO_4_, the compressive strength was 36.9 MPa. By contrast, when mixed with 2% Na_2_SO_4_, the compressive strength for the 3rd and 28th days were 22.5 and 35.8 MPa, respectively. The compressive strength still increased, but the increase was not obvious when mixed with 6% Na_2_SO_4_. Therefore, on the premise of saving costs and meeting the national standards for cementing materials, 2% Na_2_SO_4_ as the chemical activator was used.

### 3.4. The Activation Effect of Na_2_SO_4_ and Gypsum on GCM

The GCM of SSCND–cement system had better strength performance when the dosage of Na_2_SO_4_ was 2%. To explore the activation effect of Na_2_SO_4_ and gypsum collectively on the GCM of SSCND–cement system, this section was carried out using the following experiments, with the dosage of Na_2_SO_4_ kept as 2% and, to the SSCND–cement system, 2%, 4%, 6%, 8%, 10%, and 12% of gypsum as plaster was added to activate the GCM, and then the compressive strength of the samples was measured to determine the optimum ingredient ratio. The ratios of Na_2_SO_4_ and gypsum activator are shown in Table 5 (Na_2_SO_4_ chemical activator was externally added).

It can be seen from Figure 4 that when the gypsum content was 2%, the compressive strength of the GCM in the SSCND–cement system activated by the composite activator was higher than that of Na_2_SO_4_ used as the chemical activator alone. The compressive strength of the GCM at 28 days reached 41.6 MPa. When the amount of gypsum in the activator changed, the compressive strength of the GCM changed in a wide range, indicating that the composition of the composite activator had a significant impact on the properties of the gelled material. Gypsum can improve the activity of smelting slag, but if only gypsum is added as the activator, the activity of the smelting slag cannot be well activated. Only in the presence of chemical activator can a certain amount of gypsum activate the smelting slag. In addition, gypsum has the advantage of being low cost. The use of composite activator can reduce the cost of material preparation, which is cost-effective from an economic point of view. Therefore, a compound activator made of 2% gypsum and 2% sodium sulfate was selected as the optimal ratio.

### 3.5. SEM and XRD Analyses

Calcium silicate hydrate (C-S-H) and ettringite (AFt) in gel form were generated after curing the sample T3-2 for 28 days (Figure 5a). The addition of Na_2_SO_4_ resulted in formation of fine AFt crystals, which was one of the reasons for the early strength improvement. The morphology of AFt is related to the environmental alkalinity. The addition of sodium sulfate increased the environmental alkalinity, which leads to the decrease of Ca^2+^ content. AFt grows into relatively small crystals [40].

The proposed mechanism in SSCND–cement system is that, when cement is added into SSCND with water, the primary hydration reaction between minerals C_3_S (tricalcium silicate) and C_2_S (dicalcium silicate) in cement and water resulted in formation of Ca(OH)_2_:C_3_S + H_2_O 
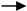
 C-S-H + Ca(OH)_2_,
C_2_S + H_2_O 
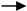
 C-S-H + Ca(OH)_2_,
and a secondary reaction between Ca(OH)_2_ and silicon dioxide, and aluminum oxide in SSCND, could solidify the heavy metals in cement clinker crystalline lattice in the process of cement kiln when solid wastes are incinerated. These elements would be compacted in the C-S-H consolidation gel or exist in small pores after cement hydration when cement is added into SSCND–cement system:m_1_ Ca(OH)_2_ +SiO_2_ +aq 
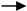
 m_1_ CaO• SiO_2_ •aq,
m_2_ Ca(OH)_2_ + Al_2_O_3_ +aq 
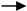
 m_2_ CaO •Al_2_O_3_ •aq.
Na_2_SO_4_, when used as the activator, can catalyze the hydration of SSCND and increase the reaction speed.

It can be seen in Figure 5b–d, the fine grains consisting of gel C-S-H were disorderly. In the massive calcium hydroxide (CH) crystals, and the hydration products in GCM in the SSCND–cement were similar to the cement gelled materials. The needle-shaped AFt crystals and flocculation C-S-H gel overlap each other to form space net structure, and populate the hardening of the phase surface, hole, crack, the unreacted dihydrate gypsum, and smelting slag particles knot together to form an integrated part, thus improving the strength of the GCM [41,42].

The XRD diffraction pattern of sample T4-2 after being cured for 28 days showed the production of calcium silicate hydrate (C-S-H), Aft, and fayalite (Figure 6), consistent with the SEM observations.

## 4. Conclusions

(1) Among the four different chemical activators, the compressive strength with NaOH as the activator meets the requirements of national standards. However, its price is high. Considering the market cost and simple operation, Na_2_SO_4_ with lower price and better activation effect was selected as the activator.

(2) When the dosage of Na_2_SO_4_ is 2%, 3%, and 6%, the compressive strength in the later stage reaches the highest and the strength change is not obvious. Considering the market cost, 2% Na_2_SO_4_ was selected to evaluate the addition amount of chemical activator for the GCM of the SSCND–cement system.

(3) It was found that the use of composite activator could optimize or improve the compressive strength of the gelled material, and when the gypsum content in the composite activator changed, the compressive strength of the prepared GCM varied widely, and the composition of the composite activator had a significant impact on the performance of GCM. Through experiments, gypsum with a mass ratio of 2% was selected and added into the GCM of SSCND–cement system.

(4) The addition of Na_2_SO_4_ provided the necessary alkaline environment for slurry, thus generating conditions for dispersion, dissolution, and hydration of SSCND. Gypsum acted as an activator to synergize with Na_2_SO_4_ to promote the activation of slag activity, resulting in hydration of backfill and further increase of strength.

(5) The final strength of the GCM reached 41.6 MPa, which meets the national GCM standard. The dosage of SSCND reached 80%. It is expected to be able to achieve bulk utilization of China’s SSCND, if full-scale industrial production took place after the pilot tests.

## Figures and Tables

**Figure 1 materials-12-00303-f001:**
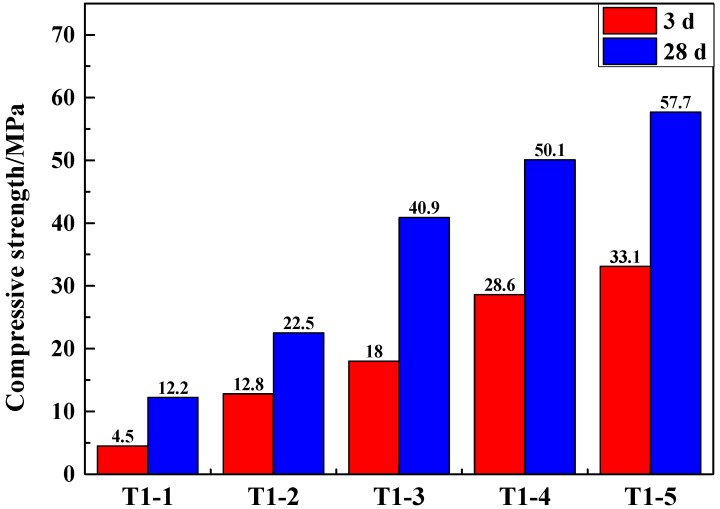
Compressive strength of samples of different SSCND–cement ratios after 3 and 28 days of curing.

**Figure 2 materials-12-00303-f002:**
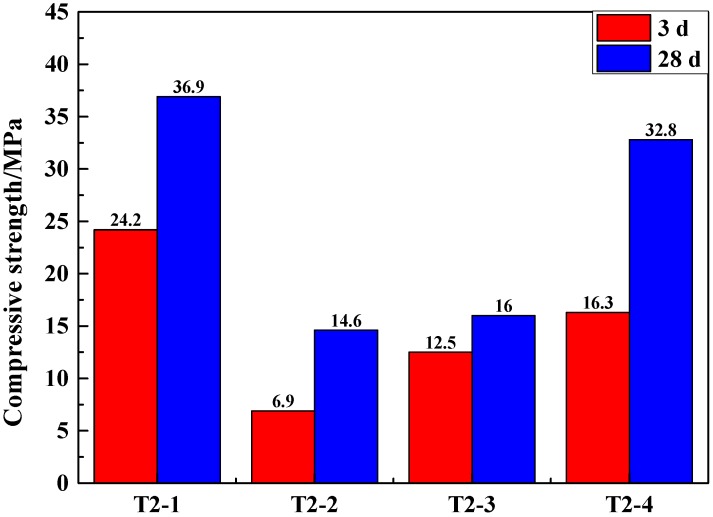
Compressive strength of the samples made of SSCND–cement at the ratio of 80:20 activated under 6% of different chemical activators after 3 and 28 days of curing.

**Figure 3 materials-12-00303-f003:**
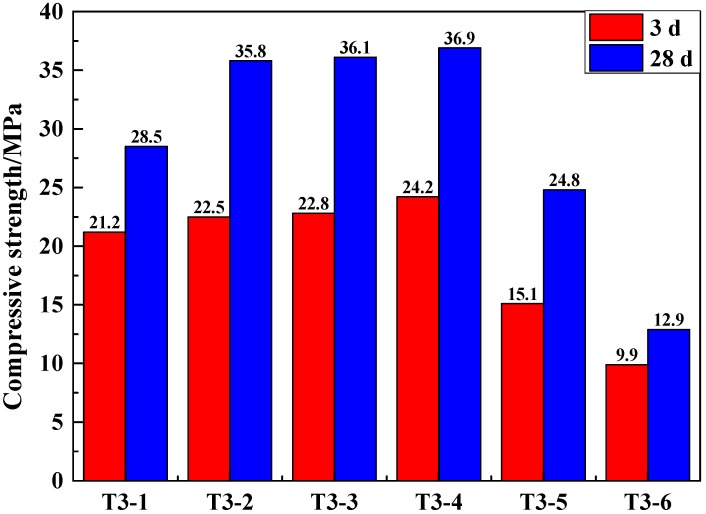
Compressive strength of samples made of SSCND–cement at the ratio of 80:20 activated under different contents of Na_2_SO_4_ after 3 and 28 days of curing.

**Figure 4 materials-12-00303-f004:**
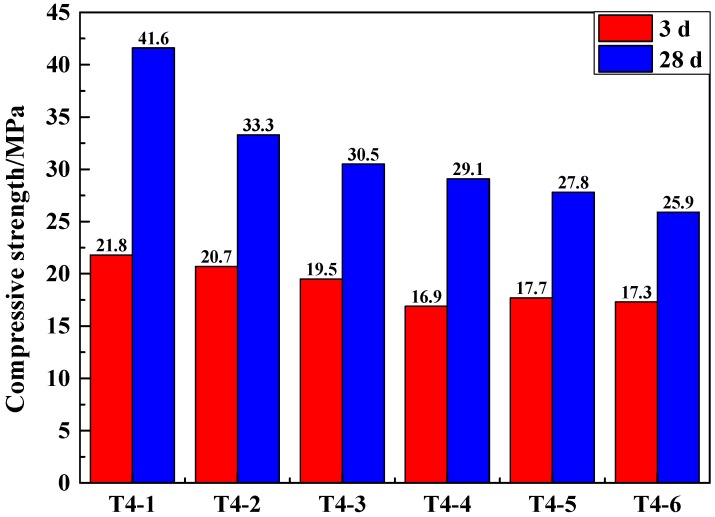
Compressive strength of the samples made of SSCND–cement at the ratio of 80:20 activated under different contents of gypsum and 2%Na_2_SO_4_ after 3 and 28 days of curing.

**Figure 5 materials-12-00303-f005:**
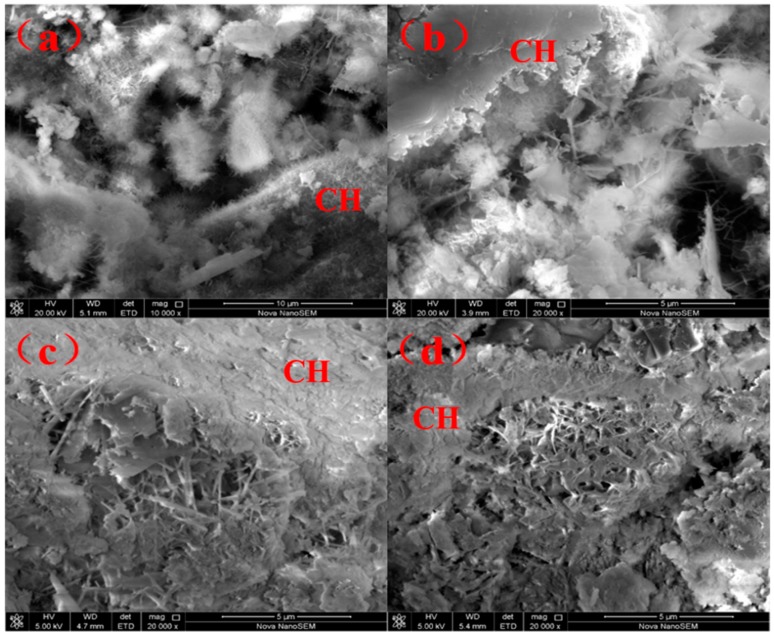
SEM images of T3-2 sample after being cured for 28 days (**a**) and of T4-1 samples for 3, 7, and 28 days of curing (**b**–**d**).

**Figure 6 materials-12-00303-f006:**
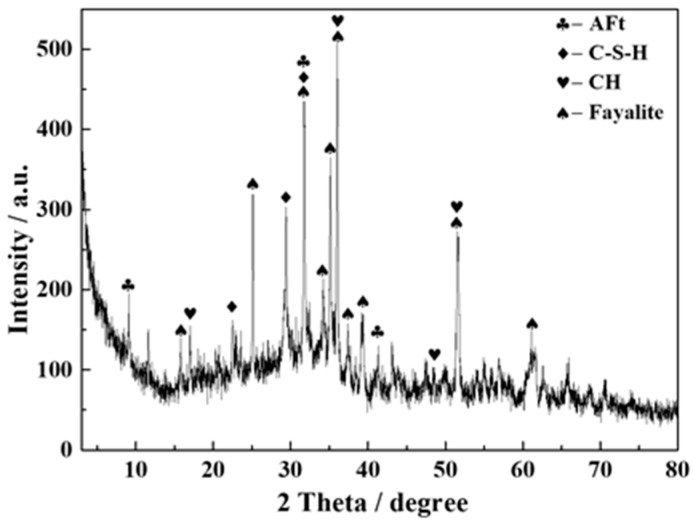
XRD pattern of T4-2 sample after being cured for 28 days.

**Table 1 materials-12-00303-t001:** Chemical composition of smelting slag of copper and nickel deposits (SSCND).

Composition	CaO	SiO_2_	Al_2_O_3_	MgO	Fe_2_O_3_	S	K_2_O	Na_2_O	P_2_O_5_	CuO	NiO
Content (%)	2.73	26.75	2.63	3.61	60.63	1.00	0.60	0.25	0.07	0.51	0.43

**Table 2 materials-12-00303-t002:** Proportion of SSCND and cement used for the preparation of gellable composite materials (GCM) (wt %).

Sample Number	SSCND	Cement
T1-1	90	10
T1-2	80	20
T1-3	70	30
T1-4	60	40
T1-5	50	50

**Table 3 materials-12-00303-t003:** The ratio of four activators added to the gelled materials (wt %).

Sample Number	SSCND	Cement	Chemical Activator
T2-1	80	20	6% Na_2_SO_4_
T2-2	80	20	6% Na_2_SiO_3_·9H_2_O
T2-3	80	20	6% Na_2_CO_3_
T2-4	80	20	6% NaOH

**Table 4 materials-12-00303-t004:** Proportion of Na_2_SO_4_ use for preparing the GCM (wt %).

Sample Number	SSCND	Cement	Na_2_SO_4_ Activator
T3-1	80	20	1%
T3-2	80	20	2%
T3-3	80	20	3%
T3-4	80	20	6%
T3-5	80	20	9%
T3-6	80	20	12%

**Table 5 materials-12-00303-t005:** Proportion of Na_2_SO_4_ and gypsum content (wt %).

Sample	SSCND–Cement = 80:20	Gypsum	Na_2_SO_4_ Activator
T4-1	98	2	2%
T4-2	96	4	2%
T4-3	94	6	2%
T4-4	92	8	2%
T4-5	90	10	2%
T4-6	88	12	2%

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
