# Peer review of "Experimental Studies on Chemical Activation of Cementitious Materials from Smelting Slag of Copper and Nickel Mine"

_materials, 2019, doi:10.3390/ma12020303_

Round 1
Reviewer 1 Report
Title: What is the chemical stimulation of cementitious materials? Abstract: What is material ratio? Between which types of materials? Make clear this. Through exciting? What is the exciting process here about? Is the exciting actually activation? Why do you call the composite materials “gelled”? What are the “other different chemical activators”? Please, remove “gelled”, and use known terms such as “composite materials”. Gelled is considered since the main reaction products in the samples are gel-based. Why then you need to write “gelled”? Sentence from the Line 15-16 is repeated in Line 18-19, please remove one. Line 15-16: The experimental results show that the chemical activators of Na2SO4, Na2SiO3, NaOH, Na2CO3 and other different chemical activators could improve the compressive strength. Line 18-19: The experiment results show that the compound chemical activator could improve the compressive strength of gelled material. Table 1, how much copper and nickel is in your slag? Where are these oxides? Table 2, would it be better if you present from lower (50-90%) to higher replacements of smelting slag of copper and nickel mine, and not other way around (90-50%) Figure 1, where is the standard deviation of strength results? Please add to the Figure. Are the presented strength results mean values? How many samples were tested for strength per mixture? What was the geometry of the moulds, you didn’t specify that in 2.2, but later in the Experiment. Please, move the geometry of the samples and curing regime to 2.2 Characterization. Did you make mortars or pastes? How did you determine which activator has the best excitation? What do you mean by excitation? Do you mean activation? 3.1. section should be moved to 2.3 Experiment Table 2. Should be moved to 2.3 Experiment Line 97 Where did you read this: Cement is a good gelling material? Line 100-101 How is 22.5 close to 32.5 MPa? Please, rewrite the sentence, this is a substantial difference. Why haven’t you made the reference sample, one with 100% cement and 100% slag? What is here the national standard, please add the reference for your national standard. Line 124 which figure? Line 126 do you have a reference for this statement? Line 128 Why the hydration rate of Na2SiO3 is slow? Line 131 do you have a reference for this statement? Line 153 Why the gypsum was used? Do you really need it? Looking the results in Figure 4, why don’t you lower the concentration of Na2SO4 due to gypsum use in Table 5, when it is shown that increase of gypsum content decreases the strength of the samples? Line 183 What is the environmental alkalinity? Line 190 where is the massive CH in the photos? Please mark it with different colour.Author Response
Title: What is the chemical stimulation of cementitious materials?
Chemically- activated cementitious materials is defined as those materials or mixture of the materials that do not have gelling properties but after chemical treatment or modification, they will show gelling propertied. The raw materials are slags of industry as well as mine tailing. After selecting and using different activating reagents, the internal structure of the materials re-organized without treatment under high temperature. As such the title of the manuscript was changes from “…. Chemical stimulation…..” to “…. Chemical activation….”.
Abstract: What is material ratio? Between which types of materials? Make clear this.
The ratio is the slag to cement to activator. They were listed in Table 5. The best ratio as T4-1.
Through exciting? What is the exciting process here about?
Sorry. We changed the wording to activation from excitation.
Is the exciting actually activation? Why do you call the composite materials “gelled”? What are the “other different chemical activators”? Please, remove “gelled”, and use known terms such as “composite materials”. Gelled is considered since the main reaction products in the samples are gel-based. Why then you need to write “gelled”?
Yes. The term was changed to activation throughout the revised manuscript. Exciting is no longer used.
We changed “gelled” to “gellable” and used GCM for gellable composite materials.
Sentence from the Line 15-16 is repeated in Line 18-19, please remove one. Line 15-16: The experimental results show that the chemical activators of Na2SO4, Na2SiO3, NaOH, Na2CO3 and other different chemical activators could improve the compressive strength. Line 18-19: The experiment results show that the compound chemical activator could improve the compressive strength of gelled material.
The repeats on lines 18-19 were removed.
Table 1, how much copper and nickel is in your slag? Where are these oxides?
The contents of Cu and Ni as oxides were added to Table 1.
Table 2, would it be better if you present from lower (50-90%) to higher replacements of smelting slag of copper and nickel mine, and not other way around (90-50%)
We partially agree with this comment. If the usage of SSCND is from low to high, the use of Cement would be from high to low. We wanted to use more SSCND for the GCM. Thus, we started with higher amount of SSCND.
Figure 1, where is the standard deviation of strength results? Please add to the Figure. Are the presented strength results mean values? How many samples were tested for strength per mixture?
We did duplicates per experimental condition. And they are very close to each other. The average of the two was use. As such standard deviation was not used.
What was the geometry of the moulds, you didn’t specify that in 2.2, but later in the Experiment. Please, move the geometry of the samples and curing regime to 2.2 Characterization.
The mold that was used had a dimension of 20 mm ´ 20 mm ´ 20 mm. It was added to section 2.2.
Did you make mortars or pastes?
The mixture had a paste-like nature and solidified in the mol.
How did you determine which activator has the best excitation? What do you mean by excitation? Do you mean activation?
We changed the word “excitation” to “activation”.
3.1. section should be moved to 2.3 Experiment Table 2. Should be moved to 2.3 Experiment
Section 2.3 is part of experimental method while section 3.1 is fro the results. We believe it is better to have Table 2 stay in section 3.1 as we are comparing the results of different experimental settings.
Line 97 Where did you read this: Cement is a good gelling material?
We added reference 35 to support this.
Line 100-101 How is 22.5 close to 32.5 MPa? Please, rewrite the sentence, this is a substantial difference.
Thank you for your comment here. 22.5 MPa is for 28 days. After a series activation reaction, the compressive strength is expected to reach 32.5 MPa.
Why haven’t you made the reference sample, one with 100% cement and 100% slag? What is here the national standard, please add the reference for your national standard.
We did the experiments with 100% cement and 100% slag. The compressive strength of 100% cement after 28 days reached 78.9 MPa. In contrast, the compressive strength of 100% slag after 28 day s had only 2.2 MPa. Data were added to the revised manuscript.
Line 124 which figure?
Fig. 2. Info added to the revised manuscript.
Line 126 do you have a reference for this statement?
Yes. Reference 36 was added to the manuscript.
Line 128 Why the hydration rate of Na2SiO3 is slow?
It was speculated that Na2SiO3will not interact with the activator and the compressive strength is on 14.6 MPa after 28 days.
Line 131 do you have a reference for this statement?
Yes. Ref 37 was added.
Line 153 Why the gypsum was used? Do you really need it? Looking the results in Figure 4, why don’t you lower the concentration of Na2SO4 due to gypsum use in Table 5, when it is shown that increase of gypsum content decreases the strength of the samples?
To add more Ca2+ to increase the reaction.
Line 183 What is the environmental alkalinity?
With the addition of cement and gypsum, the environment became alkaline.
Line 190 where is the massive CH in the photos? Please mark it with different colour.
Massive CH was labeled in the photo.

Reviewer 2 Report
For introduction, please define 'nonferrous smelting slag'. Also, please write state-of-art research on cementitious-based materials with nonferrous smelting slag. In my opinion, I cannot find why you do your experiment.
Did you make 20 mm^3 sample? I think it is quite small for your test. Why did you select it?
Main concern is that you did not show any mechanism of cement composite with nonferrous smelting slag with various chemical agent. This is very important for readers to understand your work. However, you did not mention any chemical equation to explain it.
For SEM, you only used SEM to support your higher compressive strength. However, it is not enough. Please use EDX and other technique to support your result.
Author Response
For introduction, please define 'nonferrous smelting slag'. Also, please write state-of-art research on cementitious-based materials with nonferrous smelting slag. In my opinion, I cannot find why you do your experiment.
We have made major revisions in the introduction: add the definition of 'nonferrous smelting slag' as: Nonferrous smelting slag is the waste residue in the smelting process of nonferrous metals and minerals. The implications of our study are as follows: In the process of exploiting and utilizing of mineral resource, a series of environmental problems have been arisen, such as a lot of waste rock and gob, waste of resources and security risks. The stacking of tailings and smelting slag from metallurgical process produce a serious pollution to the environment. Filling the gobs with tailings and other solid waste can not only realize the control of solid waste pollution and improve the utilization of the land, but also prevent the surface subsidence and improve the mining index. However, if tailings and other solid waste are directly used to fill the gobs, it will prone to occur oxidation and ion release in the rain, the sun and other natural stress. It is an effective means to use nonferrous smelting slag to prepare filling cementitious material for large amount of solid waste from mining and metallurgy. It had been confirmed that heavy metals could be solidified in cement clinker crystalline lattice in the process of cement kiln incinerating solid waste. These elements would be compacted in the C-S-H consolidation gel or exist in small pores after cement hydration and were not willing to release under natural conditions.
Did you make 20 mm^3 sample? I think it is quite small for your test. Why did you select it?
We did not. It is the national standard of China for the mold to measure the compressive strength.
Main concern is that you did not show any mechanism of cement composite with nonferrous smelting slag with various chemical agent. This is very important for readers to understand your work. However, you did not mention any chemical equation to explain it.
We add the following to the mechanism of cement composite with nonferrous smelting slag in section 3.5.
The proposed mechanism in SSCND-cement system is that, when cement is added into SSCND with water, the primary hydration reaction between minerals C3S (tricalcium silicate) and C2S (dicalcium silicate) in cement and water resulted in formation of Ca(OH)2:
C3S + H2O C-S-H + Ca(OH)2
C2S + H2O C-S-H + Ca(OH)2
And a secondary reaction between Ca(OH)2 and silicon dioxide, aluminum oxide in SSCND could solidfy the heavy metals in cement clinker crystalline lattice in the process of cement kiln when solid wastes are incinerated. These elements would be compacted in the C-S-H consolidation gel or exist in small pores after cement hydration when cement is added into SSCND-cement system:
m1 Ca(OH)2 +SiO2 +aq m1 CaO• SiO2 •aq
m2 Ca(OH)2 + Al2O3 +aq m2 CaO •Al2O3 •aq
Na2SO4 when used as the activator can catalyze the hydration of SSCND and increase the reaction speed.
For SEM, you only used SEM to support your higher compressive strength. However, it is not enough. Please use EDX and other technique to support your result.
The Fe content could be as high as 60%. With its high magnetism it is difficult to do an EDX measurement. We added more XRD data to support our claim.

Reviewer 3 Report
The presentation of results seems to be quite clear and the summary is sufficiently descriptive. More general information concerning the use of non-ferrous slags as components of cementitious materials. It would be better to indicate when these slags are used (not only as filling materials) and to transfer some information from the end of the contribution (authors refer their results to the previously reported by the others in) to the introduction.
English editing seems to me necessary
I am not a native speaker but I daresay that I have never seen the expressions: "gelled materials", "gelling activity" related to the hydration/binding process
Author Response
The presentation of results seems to be quite clear and the summary is sufficiently descriptive. More general information concerning the use of non-ferrous slags as components of cementitious materials. It would be better to indicate when these slags are used (not only as filling materials) and to transfer some information from the end of the contribution (authors refer their results to the previously reported by the others in) to the introduction.
English editing seems to me necessary
I am not a native speaker but I daresay that I have never seen the expressions: "gelled materials", "gelling activity" related to the hydration/binding process
We have extensively revised the manuscript and had a Professor from University of Wisconsin prove read and corrected the English of the manuscript.